# Hydrogen Bonds in Blends of Poly(*N*-isopropylacrylamide), Poly(*N*-ethylacrylamide) Homopolymers, and Carboxymethyl Cellulose

**Alberto García-Peñas** [1,2,3,*] , **Weijun Liang** [1,4] , **Saud Hashmi** [5] , **Gaurav Sharma** [1,6,7] , **Mohammad Reza Saeb** [8] **and Florian J. Stadler** [1,*]

1 College of Materials Science and Engineering, Guangdong Research Center for Interfacial Engineering of Functional Materials, Nanshan District Key Laboratory for Biopolymers and Safety Evaluation, Shenzhen Key Laboratory of Polymer Science and Technology, Lihu Campus, Shenzhen University, Shenzhen 518055, China; liangweijun2017@email.szu.edu.cn (W.L.); gaurav8777@gmail.com (G.S.)

2 College of Optoelectronic Engineering, Key Laboratory of Optoelectronic Devices and Systems of Ministry of Education and Guangdong Province, Yuehai Campus, Shenzhen University, Shenzhen 518060, China

3 Departamento de Ciencia e Ingeniería de Materiales e Ingeniería Química (IAAB), Universidad Carlos III de Madrid, 28911 Leganés, Madrid, Spain

4 Institute of Testing Technology, Institute of Jinxi Industry Group Co., Ltd., Taiyuan 030027, China

5 Department of Polymer and Petrochemical Engineering, NED University of Engineering and Technology, University Road, Karachi 75270, Pakistan; saudhashmi@cloud.neduet.edu.pk

6 International Research Centre of Nanotechnology for Himalayan Sustainability (IRCNHS), Shoolini University, Solan 173212, Himachal Pradesh, India

7 School of Life and Allied Health Sciences, Glocal University, Saharanpur 247001, India

8 Department of Polymer Technology, Faculty of Chemistry, Gdańsk University of Technology, G. Narutowicza 11/12, 80-233 Gdańsk, Poland; mohsaeb@pg.edu.pl or mrsaeb2008@gmail.com

* Correspondence: alberto.garcia.penas@uc3m.es (A.G.-P.); fjstadler@szu.edu.cn (F.J.S.)

**Abstract:** Recently, it was reported that the physical crosslinking exhibited by some biopolymers could provide multiple benefits to biomedical applications. In particular, grafting thermoresponsive polymers onto biopolymers may enhance the degradability or offer other features, as thermothickening behavior. Thus, different interactions will affect the different hydrogen bonds and interactions from the physical crosslinking of carboxymethyl cellulose, the lower critical solution temperatures (LCSTs), and the presence of the ions. This work focuses on the study of blends composed of poly(*N*-isopropylacrylamide), poly(*N*-ethylacrylamide), and carboxymethyl cellulose in water and water/methanol. The molecular features, thermoresponsive behavior, and gelation phenomena are deeply studied. The ratio defined by both homopolymers will alter the final properties and the gelation of the final structures, showing that the presence of the hydrophilic groups modifies the number and contributions of the diverse hydrogen bonds.

**Keywords:** physical crosslinking; lower critical solution temperature; cononsolvency phenomenon; hydrogen bonds

## 1. Introduction

The study of the physical crosslinking exhibited by grafting smart thermosensitive structures is growing due to the multiple benefits obtained for many applications, such as gelling polymers, materials for removal of heavy metals, or degradation agent [1–4]. Some studies were performed using carboxymethylcellulose (CMC) due to its frequent use as a viscosity modifier and stabilizer, which is used for many applications in cosmetics, food, or pharmaceutical industries. It has been observed that the viscosity of CMC increased when glycerin was added [5]. Later, the phenomena of gelation and changes in viscosity were explained by the physical crosslinking of hydrogen bonds [6], which were studied in CMC solutions prepared by mixtures of glycol/water and poly(vinyl alcohol)/water. [1,7,8].

Intrinsic viscosity results of solutions of CMC in ethylene glycol/water and methanol/water mixtures were also analyzed [9,10].

Grafting thermoresponsive polymers onto CMC led to thermothickening behavior [11–14] or better degradability properties [15]. Furthermore, these structures showed good characteristics as nanocomposites for treating wastewater, especially for removing metal ions [16]. Nevertheless, some problems were found due to the high salinity, which promotes the precipitation of the polymeric chains. Using thermoresponsive polymers like poly(*N*-ethyl acrylamide) (pNEAM), where the lower critical solution temperature is above 40 °C, could diminish this effect.

The thermoresponsive polymers can undergo the cononsolvency phenomenon in some mixtures of solvents, i.e., a volume phase transition is observed. In general, these studies were focused on poly(*N*-isopropylacrylamide) (pNIPAM), where various parameters as density, hydrophobic interactions, or chain lengths were associated with the cononsolvency phenomenon [17–20]. The pNIPAM testified this effect in water mixtures with methanol or ethanol, among others [21–30]. The effect of monomer compositions was studied in copolymers based on *N*-isopropylacrylamide (NIPAM) and *N*-ethyl acrylamide (NEAM), where it was observed that hydrophobic polymers show a better cononsolvency behavior than hydrophilic ones [17,31].

In general, these manuscripts studied specific isolated changes of hydrogen bonds related to the novel physical crosslinking missing other important effects which can take places, such as the cononsolvency or the well-known lower critical solution temperature (LCST). For that reason, a study about all these phenomena related to the hydrogen bonds in solution could be significant to extend the spectra of applications and knowing the limitations of these new structures. Many questions must be addressed: What is the competition between the different hydrogen bonds associated with the lower critical solution temperature, the physical crosslinking, and the cononsolvency phenomenon? Can we know the different contributions? Are there some restrictions associated with the ratios between homopolymers? What is happening if we add another solvent over water?

This work tries to identify the contributions of these parameters over the phase transition temperature and the gelation in blends based on poly(*N*-isopropylacrylamide), poly(*N*-ethylacrylamide) homopolymers and carboxymethyl cellulose in water, and mixtures of water and methanol. The thermoresponsive behavior and properties were studied by UV-vis spectroscopy, dynamic light scattering, and rheology. The results were compared with the molecular characteristics of structures.

## 2. Experimental

### 2.1. Materials

The monomers were purified before synthesis, where *N*-isopropylacrylamide (NIPAM, 98%, Xiya Reagent, Shandong, China) was recrystallized using n-hexane and toluene (9:1), and *N*-ethylacrylamide (NEAM, 99%, Aldrich, St. Louis, MO, USA) through an alumina column. Azobisisobutyronitrile (AIBN, Aldrich, St. Louis, MO, USA) was selected as the initiator and was recrystallized. Sodium carboxymethyl cellulose (CMC, Macklin, Shanghai, China), deionized water (Millipore Elix 5$^{UV}$ Millipore Corporation, Billerica, MA, USA), diethyl ether (99.5%, Lingfeng, Shanghai, China), and methanol (MeOH, 99.5%, Damao chemical reagent, Tianjin, China) were used as received. *N*,*N*-Dimethylformamide (DMF, 99.5%, Macklin, Shanghai, China) was treated with nitrogen gas to remove the oxygen of the solvent.

### 2.2. Synthesis and Mixtures of Homopolymers

The homopolymers were synthesized following the next steps. First, poly (*N*-isopropylacrylamide) (pNIPAM) and poly(*N*-ethyl acrylamide) (pNEAM) were independently prepared using specific amounts (Table 1) of monomers (NIPAM and NEAM, respectively) and initiator (AIBN) and using DMF as a solvent. Inert conditions were established for reactants placed inside of two Schlenk tubes. Then, both Schlenk tubes

were placed in a thermostatic bath at 75 °C for 18 h for keeping the same conditions. Both independent reactions were stopped by cryogenization using liquid nitrogen.

**Table 1.** Preparation of homopolymers.

| Name | NIPAM (mol) | NEAM (mol) | AIBN (mol) | DMF (mL) |
|---|---|---|---|---|
| pNIPAM | 0.022 | – | $6 \times 10^{-4}$ | 15 |
| pNEAM | – | 0.028 | $6 \times 10^{-4}$ | 14 |

Then, different amounts of the prepared pNIPAM and pNEAM were mixed, varying the ratio between homopolymers as indicated in Table 2. The blends were precipitated in diethyl ether, purified twice, and dried in vacuo at room temperature.

**Table 2.** Composition of the mixtures of homopolymers.

| Name | pNIPAM (%) | pNEAM (%) |
|---|---|---|
| mNEAM$_{25}$ | 75 | 25 |
| mNEAM$_{70}$ | 30 | 70 |
| mNEAM$_{90}$ | 10 | 90 |

The blends of both homopolymers were designated mNEAM, where m means mixture, followed by the NEAM-content. For instance, a polymer whose content is around 90% of NEAM, will be denominated as mNEAM$_{90}$. Then, polymer water solutions were prepared for analyzing the thermoresponsive behavior as a reference before preparing the new blends with CMC. The samples were placed inside a refrigerator (4 °C) for homogenization.

### 2.3. Preparation of Blends Based on Thermoresponsive Polymers and CMC

The blends, composed of poly(*N*-isopropylacrylamide) (pNIPAM), poly(*N*-ethyl acrylamide) (pNEAM), and CMC, were prepared in water and water/methanol at 0 °C. A better homogenization of the samples was carried out through two steps. First, the polymers were dissolved in water and stored in the refrigerator for 24 h, and subsequently, the CMC was added under stirring according to the proportions of Table 3. In the case of samples prepared in mixtures of methanol and water, the methanol was incorporated in the last step at low temperature (below LCST) using an ice bath. Finally, all the samples were stored in the refrigerator for 24 h before analysis.

**Table 3.** Composition of blends composed of polymers and CMC in water and water/methanol mixtures.

| Name | Polymer Concentration (wt.%) | CMC Concentration (wt.%) | Water: MeOH (Ratio) |
|---|---|---|---|
| mNEAM$_{25}$CMC | 10 | 2.5 | 100:0 |
| mNEAM$_{25}$CMC$_{MeOH}$ | 10 | 2.5 | 50:50 |
| mNEAM$_{70}$CMC | 10 | 2.5 | 100:0 |
| mNEAM$_{70}$CMC$_{MeOH}$ | 10 | 2.5 | 50:50 |
| mNEAM$_{90}$CMC | 10 | 2.5 | 100:0 |
| mNEAM$_{90}$CMC$_{MeOH}$ | 10 | 2.5 | 50:50 |

The blends were called the blends of homopolymers, followed by CMC, whose molar mass is around 250,000 g/mol and the degree of substitution of the Na$^+$ is 0.9. The MeOH was added to names for the blends performed in water/methanol mixtures. For instance, a blend composed of mNEAM$_{90}$ and CMC in water will be denominated as mNEAM$_{90}$CMC, while the same one in water/methanol will be identified as mNEAM$_{90}$CMC$_{MeOH}$. The list of the obtained samples is shown in Table 3.

## 2.4. Molecular Characterization

The structure of the polymers was analyzed using nuclear magnetic resonance and gel permeation chromatography. The proton nuclear magnetic resonance spectra ($^1$H-NMR) were performed in an AVANCE III 600 MHz spectrometer (Bruker, Karlsruhe, Germany) at 25 °C using deuterated $D_2O$. The molecular weights and distributions were studied using size exclusion chromatography (Waters e2695 Separation Module, Waters, Taunton, MA, USA) at 35 °C, using tetrahydrofuran as solvent (1 mL/min). The materials were also analyzed by Fourier transform infrared spectrometer (FTIR, Nicolet 6700, Waltham, MA, USA), samples were incorporated in tablets of potassium bromide.

## 2.5. Properties

The thermoresponsive behavior was analyzed using a UV-vis spectrophotometer (PerkinElmer UV/VIS Lambda 365) linked to a temperature controller (PerkinElmer 365K7010302, Waltham, MA, USA). The samples (1 wt.%) were tested from 20 to 95 °C using a heating rate of 1 °C/min at 400 nm.

Dynamic light scattering (Zetasizer Nano ZSP, Malvern Panalytical, Malvern, U.K.) analysis was selected as another way for detecting the changes associated with the phase transition temperature. The experiments were carried out between 20 to 92 °C, using transparent cuvettes (quartz, Purshee, Beijing, China) for the blends (0.03 wt.%) and a heating rate of 2 °C/min.

Using an Anton Paar MCR 302 rheometer (Graz, Austria), the rheological measurements were performed to test for the LCST-behavior and the viscosity function. The analysis of LCST-behavior was obtained by temperature dependence measurements, which were evaluated using a heating rate q = 1 K min$^{-1}$, angular frequency $\omega$ = 1 rad s$^{-1}$, and deformation amplitude $\gamma_0$= 1% (linear conditions) in humidity saturated atmosphere. A temperature range between 3 and 95 °C was tested. Dynamic frequency sweeps (DFS) tests were completed at angular frequencies ranging from 100 to 0.1 rad/s and using a shear deformation amplitude $\gamma_0$ of 1%. On the contrary, dynamic strain sweeps (DSS) had a fixed angular frequency of 1 rad/s and shear deformation amplitude between 0.01% to 1%. Both measurements were carried out at 3 °C (below LCST), and the samples were isolated from the environment using a low viscous paraffin oil to avoid the volatilization of the solvent.

## 3. Results and Discussion

The molecular characterization of obtained polymers was carried out through gel permeation chromatography and proton nuclear magnetic resonance. Figure 1a exhibits the GPC-data for mNEAM$_{90}$, where both peaks can be clearly detected for pNIPAM and pNEAM; the ratios of these two peaks belonging to the individual homopolymers.

The molecular weight ($M_w$) and polydispersity indices (PDI—the ratio of weight $M_w$ and number average molar mass $M_n$) of pNIPAM and pNEAM homopolymers were estimated from GPC-data. Their number average molar masses were $M_n$ = 29,000 g/mol (PDI = 1.52) and 3500 g/mol (PDI = 1.05), respectively, based on PS-standards. The differences between molecular weights could be associated with different effects associated with the chemical kinetics. The preparation of the different homopolymers, and the subsequent mixture, provided high homogeneity in the resulting structures. The homogeneity of the blends was assured because the same homopolymers were used for their preparation.

The FTIR-spectra (Figure 1b) shows similar structures for all the blends of homopolymers, as could be expected from the synthetic route, showing an excellent equivalence between samples.

The final structure of the pure polymers and blends was determined by comparing the spectra of proton nuclear magnetic resonance (Figure 1c). NMR-data can provide important information about the microstructure of the resulting polymers. Deuterated water ($D_2O$), used as a solvent, shows a clear peak at 4.8 ppm [32]. The similarities between spectra indicate that structures are similar and can exclusively be attributed to differences in blend

composition, as Figure 1c shows by the isolated peaks associated with the methine of pNIPAM and the methylene of pNEAM placed at 3.9 ppm and 3.2 ppm, respectively [32].

The variation between properties must be predominantly linked to the ratio between homopolymers (Table 4). The relative composition between polymeric chains was also observed by $^1$H-NMR, showing a good correspondence with the physical blends. It was estimated by the relationship between the methine of pure pNIPAM and the methylene of pNEAM for the different blends.

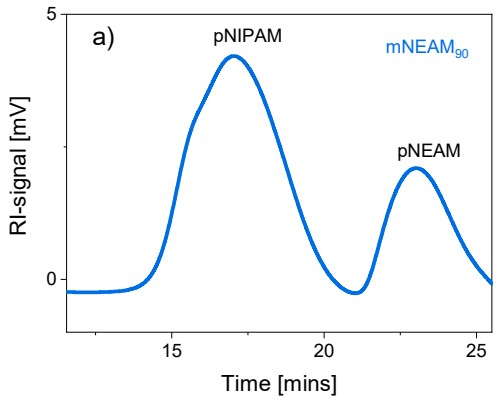

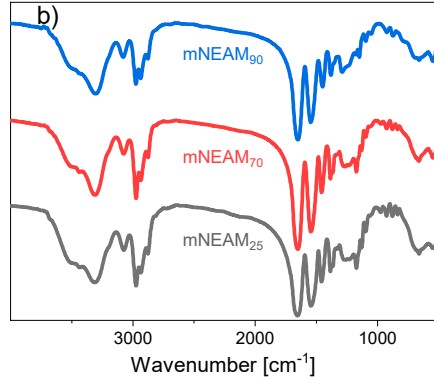

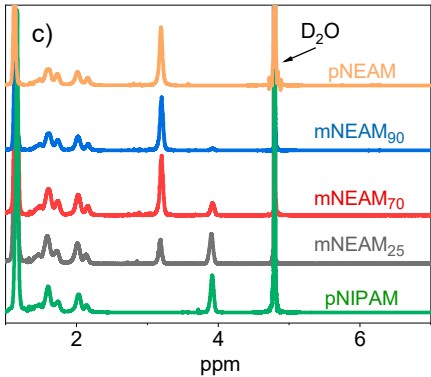

**Figure 1.** Molecular characterization of polymers and blends obtained by GPC (**a**), FTIR-spectra (**b**), and $^1$H-NMR (**c**).

The phase transition temperature associated with the LCST was studied for all the original mixtures of homopolymers. This study could explain the effects of the molecular features on the final LCST through the sensibility obtained by rheology. In this context, the

verification of the rheological data was carried out using UV-vis spectroscopy and dynamic light scattering, where other important information can also be achieved.

**Table 4.** Ratio between homopolymers for different blends.

| Name | NIPAM [mol.%] | NEAM [mol.%] |
|---|---|---|
| $mNEAM_{90}$ | 8 | 92 |
| $mNEAM_{70}$ | 21 | 69 |
| $mNEAM_{25}$ | 75 | 25 |

Figure 2a exhibits the data derived from UV-vis spectroscopy, where two phase transition temperatures can be detected for polymeric water solutions of mixtures of homopolymers (1 wt.%). The transition placed at a lower temperature can be identified as the LCST of pNIPAM because the LCST related to pNIPAM is defined at around 32 °C [33,34]. On the other hand, the second LCST placed at higher temperatures is associated with pNEAM, whose values were previously found to be between 62 and 82 °C depending on the measurement method and molar mass [35–37]. Both LCSTs can be identified for all mixtures with the transmission between the LCSTs scaling with the mixing proportion. However, the curve of $mNEAM_{70}$ was very close to $mNEAM_{25}$, suggesting a nonlinear dependence due to the nature of the transmission reduction. It could be understood by considering this to be the consequence of a threshold-type behavior, i.e., the reduction of transmission follows a sigmoidal function with the switching concentration being between 90 and 70% pNIPAM. Furthermore, $mNEAM_{90}$ showed an increase in transmittance above 40 °C before decreasing again due to the LCST of pNEAM. This could be understood as the consequence of the pNIPAM chains clustering into bigger, less light scattering clusters that could also move out of the observation range as they might sink downwards in the vial owing to a higher density.

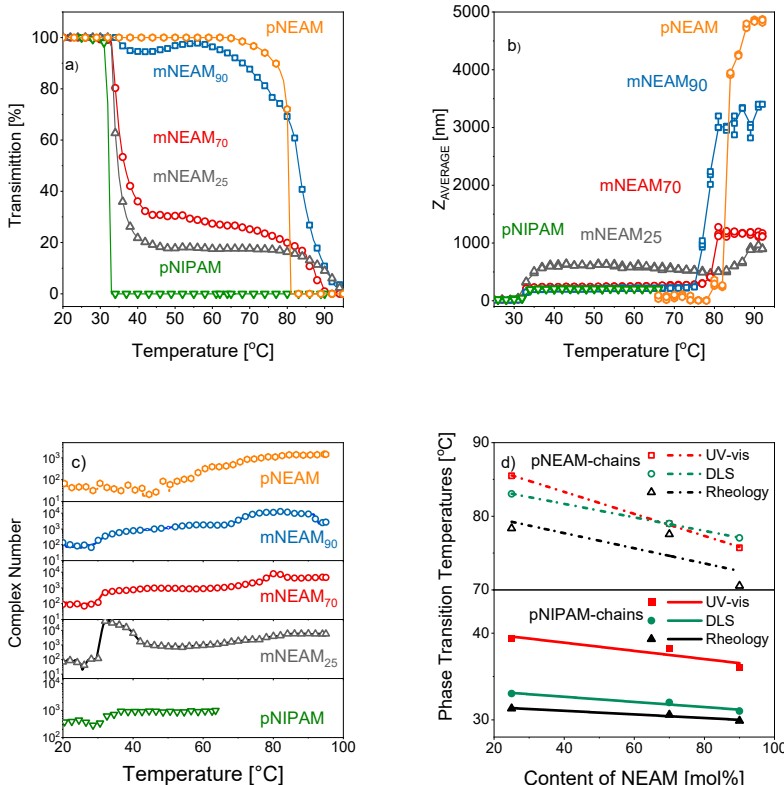

**Figure 2.** Phase transition temperatures analyzed for all the blends composed of pure polymers by UV-vis spectroscopy [1 wt.%] (**a**), dynamic light scattering [0.03 wt.%] (**b**), and rheology [10 wt.%] (**c**). LCSTs of pNEAM and pNIPAM vs. pNEAM-content (**d**).

The thermoresponsive behavior was also studied by dynamic light scattering (Figure 2b) using a polymeric concentration in water of 0.03 wt.%. Again, both transitions are identified through the ratio between polymeric chains associated with pNIPAM and pNEAM. Furthermore, the trend and temperatures followed the behavior described by UV-vis spectroscopy [38,39]. The LCST of pNIPAM leads to an increase of the Z-average particle size ($Z_{AVERAGE}$), where mNEAM$_{90}$ and mNEAM$_{70}$ show almost identical values between 35 and 75 °C (above the LCST of pNIPAM but below the LCST of pNEAM), suggesting that the pNIPAM colloidal particles are sufficiently small to remain non-percolated. Upon reaching the LCST of pNEAM at 75–80 °C, the high fraction of pNEAM in mNEAM$_{90}$ and mNEAM$_{70}$ leads to a rapid increase of Z-average particle size due to percolation of the formation of a hydrophobic polymer chain network. mNEAM$_{25}$, in contrast, shows a much higher Z-average particle size above the LCST of pNIPAM and only a moderate further increase at the LCST of pNEAM, which is the consequence of the much higher pNIPAM content. Overall, the results suggest that the hydrophobic clusters formed by pNIPAM are denser and therefore also smaller than those formed by pNEAM, making them appear bigger in light scattering.

Figure 2c shows the rheological analysis for all the blends. The transitions are evident as an increase of the magnitude of complex viscosity $|\eta^*|(T)$ confirms the previous results of UV-vis spectroscopy and dynamic light scattering. These results can be directly associated with the microstructure of the polymers studied [33,34,40]. Only two transitions are detected, as it could be expected from the preparation route, i.e., as the mixtures are not copolymers, there are no other effects associated with random sections composed by NIPAM and NEAM [33,34,40]. On the other hand, the abrupt fall exhibited by mNEAM$_{25}$ after the first phase transition temperature can be explained through artifacts, which were discussed in detail elsewhere [41]. It is well established that the LCST of pNIPAM is rather pronounced [41] 75% of mNEAM$_{25}$ are in pNIPAM-chains, which, therefore, will dominate the rheological behavior.

Valuable information can be achieved by comparing LCSTs estimated by UV-vis spectroscopy, dynamic light scattering, and rheology of both constituents (Figure 2d). The cloud temperatures (CP) were calculated at 50% of the transmittance value change, the phase transition temperature (PT) from dynamic light scattering was estimated at 50% of the collapse transition, and the transition temperature ($LCST_R$) derived from rheology was elected as the half-height of the increase in $|\eta^*|$. Figure 2d exhibits all the LCSTs calculated by these methods as a function of the ratio between pNIPAM and pNEAM. The highest phase transition temperatures show a clear trend for all the techniques, i.e., the phase transition temperature decreases when pNEAM-ratio rises.

The overall trend is that increasing the pNEAM-content (and decreasing the pNIPAM content) leads to a decrease of the LCST of pNIPAM and pNEAM by 3.4 K and 9.8 K, respectively. It is well established that the LCST of pNEAM varied in a much larger range than its pNIPAM-counterpart, although mostly that was attributed to the molar mass only [33]. In general, increasing the pNEAM content should increase the LCST in random copolymers, as was explained before [17], but these systems are composed of simple mixtures of pNIPAM and pNEAM homopolymers. Thus, the collapse of the pNIPAM chains could partially restrict the hydrophobic interactions between other polymeric chains. If more pNIPAM chains are collapsed, fewer pNEAM chains could interact between them and other polymeric chains, so the LCST would be higher. Nevertheless, if more pNEAM-chains interact between them and surrounding polymeric chains, the LCST will decrease due to a lower content of pNIPAM-chains contracted.

The polymer concentration did not show a clear trend as measurements were carried out through different techniques, as it can be deduced from the phase transition temperatures of rheology (10 wt.%), light scattering (0.03 wt.%), and UV-spectroscopy (1 wt.%). Nevertheless, the error derived from the measurements should be considered, and other parameters could also be involved. This finding did not invalidate the idea of a small con-

centration dependence, e.g., determining the LCST by DSC, as the effect of concentration is small but systematic [33,40].

Possible reasons for the concentration dependence of the LCST of pNEAM could be the presence of globules of insoluble pNIPAM chains that, due to their small size at low pNIPAM-contents, could lead to an LCST at a lower temperature than at higher pNIPAM-contents, which should have larger insoluble clusters. The dependence of the LCST of pNIPAM on its content could be due to the Hofmeister arguments [42], making pNEAM behave chaotropic. One might argue the same for the role of pNIPAM for the pNEAM-LCST, but unlike in the former case, pNIPAM was hydrophobic and phase-separated, so it could not behave like a dissolved component. These phenomena could also be understood in terms of loss of order in the water molecule arrangement around the hydrophobic chains and intramolecular hydrophobic interactions between isopropyl and ethyl groups dominating at lower pNIPAm concentrations or less hydrogen bonding moieties that are responsible for lowering LCST [43]. At higher concentrations of pNIPAM, the mobility and the availability of hydrogen bonding moieties are higher. Ultimately, the entropic mixing effect is also smaller because of larger particles; hence, it does not allow better intermolecular hydrophobic interactions, resulting in higher LCST [44].

The blends composed of polymers and CMC were prepared in water and mixtures of water and methanol, as explained in the Experimental Part. The blends, performed in water, were transparent and showed a slightly higher density than normal polymeric solutions. Appreciable changes between the constituent contents could not be observed with the naked eye. Nevertheless, the situation changed when the samples were prepared in mixtures of methanol and water. In that case, sample density suddenly increases, and the systems become opaque and whitish due to cononsolvency, which varies depending on the ratio between homopolymers. Furthermore, these changes could also be explained through previous works where authors described an increase of viscosity in Na-CMC solutions prepared in mixtures of ethylene glycol/water and methanol/water, which was related to the physical crosslinking of CMC [1,8].

In addition, the white color could be explained by the cononsolvency phenomenon, where a mixture of solvents behaves like a poor solvent at specific ratios but separately are good solvents for the same polymer [17,21–23]. The gelation produced by the physical crosslinking of CMC seemed disrupted by the presence of the polymers and probably by the cononsolvency phenomenon.

The high density and turbidity of the obtained blends in methanol/water reduce the number of techniques able to detect the changes in the phase transition temperature to non-optical techniques. Rheology can get an insight into these samples, as the sample transparency does not influence the mechanical properties. In many cases, they are related through various processes influencing both properties simultaneously.

The blends prepared in water (Figure 3) clearly show both transitions observed in the polymeric water solutions, but both phase transition temperatures decrease due to the presence of CMC. Higher contents of ions induce a lower phase transition temperature [11]. Furthermore, the LCST is very similar independently of the ratio of pNIPAM and pNEAM, which is a clear indicator of the independence between different polymeric chains, as for a random-copolymer, the LCST-temperatures would vary systematically with comonomer content [17,37,45,46].

The presence of CMC decreases the LCST, as the introduction could expect it of ions inside the polymeric solution in accordance with Hofmeister series arguments [42]. Thermothickening behavior was not observed, i.e., the viscosity did not rise when the temperature increases [47].

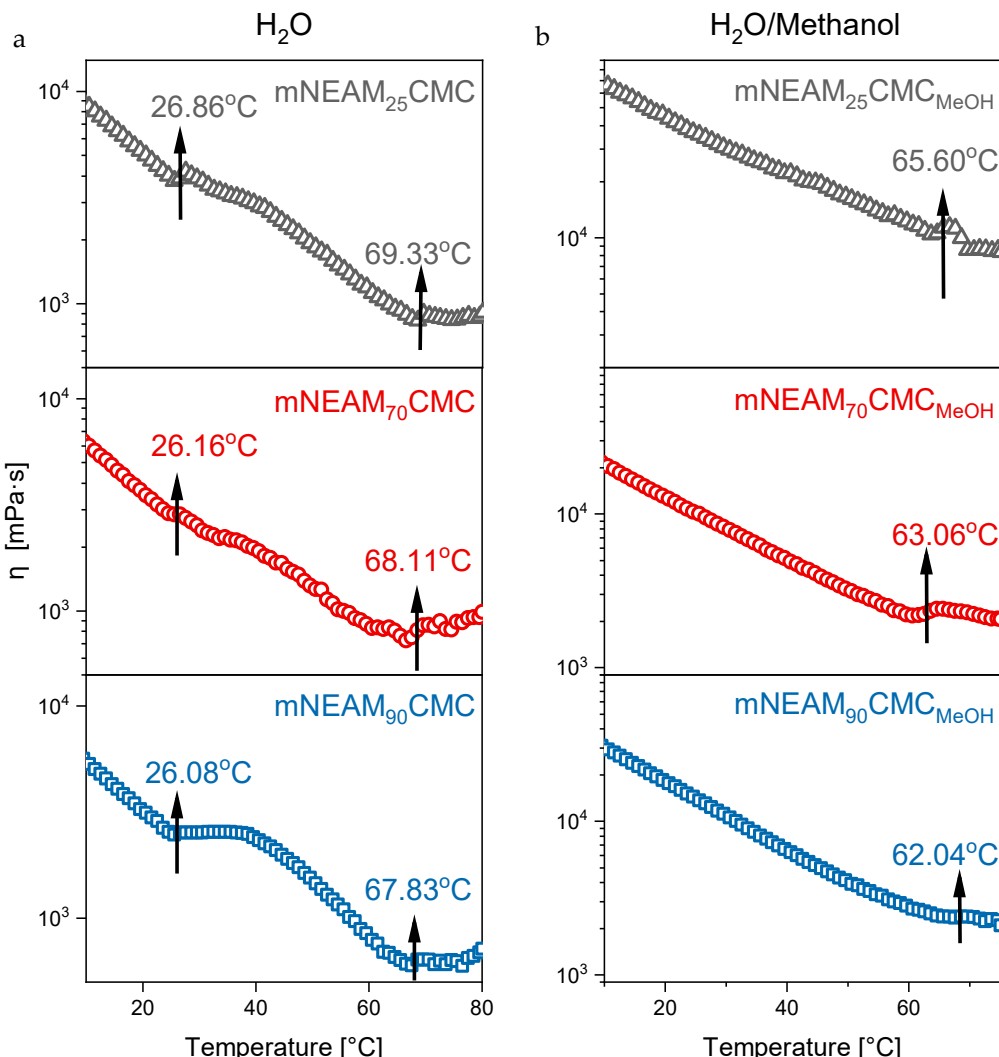

**Figure 3.** Complex viscosities as a function of the temperature of blends prepared in water (**a**) and water/methanol (**b**).

The samples performed in methanol and water (Figure 3) exhibit unusual behavior, as the first LCST (associated with pNIPAM-chains), cannot be observed. The cononsolvency could explain this effect because pNIPAM shows no or only a weak LCST in mixtures of methanol and water (1:1) once the methanol content exceeds ca. 30% [17,37,45,46,48]. This fact could be attributed to the hydrophobic behavior of the pNIPAM polymeric chains. Nevertheless, the LCST of pNEAM can be clearly detected, as hydrophobic polymers display a better cononsolvency than hydrophilic polymers [17]. In addition, the phase transition temperature is ca. 4–5 K lower than in aqueous solutions in the presence of MeOH [17]. This fact could not be distinguished for mixtures of random copolymers based on NIPAM and NEAM, as a single phase transition temperature associated with LCST was detected related to both comonomers [31].

The phase transition temperatures of the blends were estimated, as explained before. Figure 4a exhibits the phase transition temperatures of pNIPAM-chains, where the blends show a small decrease in the LCST-values due to the presence of ions [41]. Like for the samples without CMC, a slight decrease in phase transition temperature could be detected when the content of pNEAM-chains increases. A molar mass or structure (e.g., tacticity) influence can be discarded as the samples are composed of the same homopolymers at different mixing ratios. In this sense, the molar mass could play an important role as the molar mass of pNEAM ($M_n$ = 3500 g/mol) was rather lower than pNIPAM

($M_n$ = 29,000 g/mol), i.e., if the pNEAM amount increased the interactions between polymeric chains consequently raised. This behavior is also exposed by the highest phase transition temperatures of pNEAM-chains in Figure 4b, where clearly, the amount of interactions associated with pNEAM decreases the LCST for $mNEAM_x$.

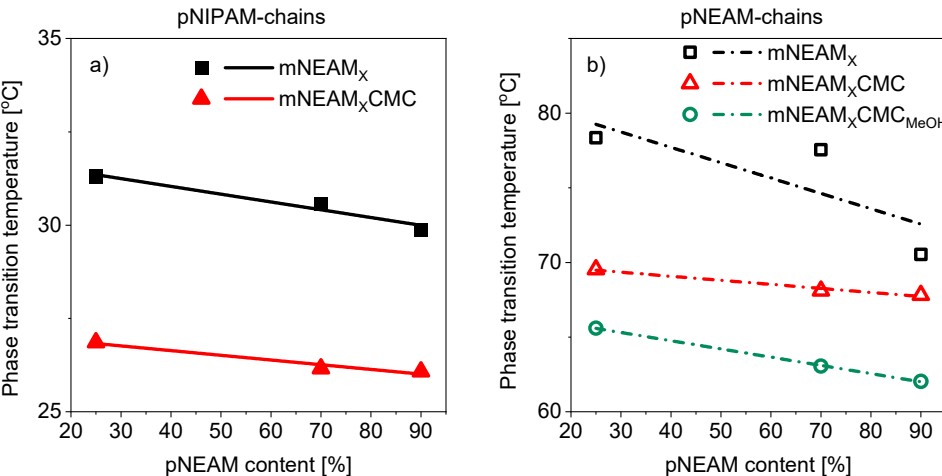

**Figure 4.** Phase transitions temperatures of polymers and blends associated with pNIPAM-chains (**a**) pNEAM-chains (**b**).

The collapse of the pNIPAM-chains after the first phase transition temperature associated with LCST could be involved in the trend of LCST described by $mNEAM_x$ samples when pNEAM content rises. The collapse of the pNIPAM-chains could decrease the interactions between polymeric chains of pNEAM, specifically when more pNIPAM chains are inside the solution. Nevertheless, if the content of pNEAM increases regarding pNIPAM, the highest LCST could decrease as interactions between chains rise. In this case, that effect could justify the change between $mNEAM_{25}$ and $mNEAM_{90}$, where the collapse of the pNIPAM-chains shows a variation of 7 K between both samples.

The presence of CMC reduces the LCST due presence of ions. This fact is also supported by a lower slope of the $mNEAM_xCMC$ samples compared to $mNEAM_x$ mixtures. The decrease of the phase transition temperature associated with LCST due to the presence of CMC showed a minimum for $mNEAM_{25}CMC$, which value was lower than 10 K regarding $mNEAM_{25}$. Similar transition temperatures were observed for $mNEAM_{70}CMC$ and $mNEAM_{90}CMC$, exhibiting the strong effect of ions over LCST regarding other parameters as the ratio between both homopolymers.

The subsequent addition of methanol decreased the LCST similarly to the addition of CMC because a higher content of ions was inside the solution. The lack of sensitivity associated with the LCST of pNIPAM, probably promoted by the cononsolvency effect, together with the impact of the hydrogen bonds of the physical crosslinking between CMC and methanol, would affect the LCST-behavior. Consequently, a deep study should be carried out.

The gelation was studied through linear viscoelasticity analysis using small amplitude oscillatory shear (SAOS) for all the blends performed in water (a) and water/methanol (b) at 3 °C (Figure 5).

The aqueous samples showed the typical behavior of a polymer solution or melt in the terminal regime. One might argue that $mNEAM_{25}CMC$ and $mNEAM_{70}CMC$ show a small upturn of the storage modulus towards low angular frequency $\omega$, suggesting some kind of slow relaxation process being present. This slow relaxation process could be related to the bridging of CMC molecules by pNIPAM, as it disappeared at the lowest pNIPAM-contents, which could be explained through the physical crosslinking of CMC. In all other aspects, the data looked very similar, being typical for the terminal regime of a polymer solution. This is logical considering that only the blend ratio of pNIPAM and

pNEAM was changed. The CMC was the same for all the experiments, and exclusively, changes in gelation could be associated with the polydispersity and ratio between pNIPAM and pNEAM homopolymers. Then, the differences could be related to the interactions between pNIPAM, pNEAM, and CMC, where competition between different hydrogen bonding is happening [1]. When disregarding the difference in thermoresponsive behavior, which could be neglected here, as 3 °C was significantly below the LCST, the somewhat higher moduli of mNEAM$_{25}$CMC are the consequence of this material's higher molar mass, shifting the terminal relaxation, thereby increasing the moduli.

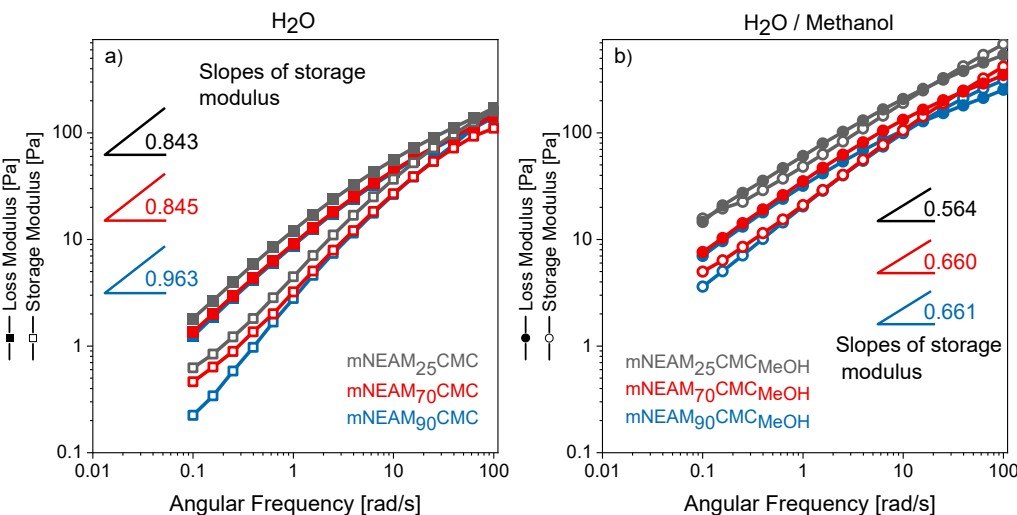

**Figure 5.** Mechanical spectra for blends prepared in water (**a**) and water/methanol (**b**).

Upon adding methanol, the moduli increased significantly, and the behavior of the materials bore some similarity with that of a critical gel, i.e., a more or less frequency-independent phase angle δ of ≈45° [49]. The formation of any gel was accompanied by either jamming or some kind of network formation, which could be associated with the physical crosslinking between the CMC and methanol. In this case, water-methanol mixtures led to a dramatically reduced solubility of the homopolymers in the blends, as discussed before, owing to cononsolvency. When looking at the data in more detail, it becomes obvious that the higher the pNEAM-content, the less the rheological data differs from the behavior of a polymer solution (see, e.g., Figure 5a). Higher pNEAM-contents lead to the presence of a slow process, which is a reduction of G′-slope at low frequencies. Considering the cononsolvency, this process is most likely caused by some insoluble fractions of the homopolymers, particularly the pNEAM, in which more or less hairy colloidal particles interact. In comparison to the unentangled polymer solution, such interactions led to an increase in viscosity overall.

The gelation of CMC through physical crosslinking seems evident in Figure 5b regarding Figure 5a. All the trends are similar but slightly lower than samples prepared in water due to the hydrogen bonds associated with CMC in mixtures of propylene glycol and water [1]. On the other hand, the cononsolvency phenomenon is also presented in the pNIPAM-chains. It can be deduced through the milky samples obtained and the absence of LCST-signal in mixtures of water/methanol (right side of Figure 3). Interaction between hydrogen bonds associated with pNEAM-chains could be minimized due to the cononsolvency behavior that could affect the pNEAM too, and the LCST related to pNEAM-chains which occurred at really high temperatures.

Thus, the partial gelation due to hydrogen bonds of CMC seems slightly affected by the LCST-phenomenon of the blends of homopolymers, i.e., the cononsolvency effect observed in solutions prepared in water/methanol mixtures could restrict the hydrogen bonds of the physical crosslinking depending on the ratio between pNIPAM and pNEAM. On the other hand, the pNIPAM could induce the formation of clusters that isolate the

thermoresponsive blend of homopolymers regarding the CMC. Similar behavior was exhibited by random copolymers composed of the same monomers and CMC, but a clear explanation was not achieved due to different copolymers were used, and other kinds of parameters were involved as polydispersity or molecular weight [31].

Figure 6 shows the responses of the complex viscosity $|\eta^*|$ as a function of shear strain $\gamma_0$ for the different blends prepared in water and water/methanol mixtures. The samples performed in water show lower complex viscosities than other blends performed in water/methanol mixtures, which agrees with the frequency dependence (Figure 5). The nonlinearity limit, defined by the deformation $\gamma_0$, at which the material functions (here $|\eta^*|(\gamma_0)$) showed a 5% change from the linear viscoelastic behavior ($\gamma_0 \rightarrow 0$). For polymer solutions and melts and colloidal polymer gels, whose colloidal nature is not too dominating, the nonlinearity limit is usually around $\gamma_0 = 30\%$ [50–53]. For any system with clear phase boundaries, however, the nonlinearity limit is reduced significantly. Typical colloidal systems have a linearity limit $\gamma_0$ around 1%, but it can also be significantly lower [50]. The nonlinearity limits for the aqueous system is ca. 60%, suggesting that they behave like classical solutions, while for the methanol-water systems, nonlinearity limits around 6% were found, which indicated that the system has some colloidal behaviors, which agreed to the cononsolvency discussion before, as a partial phase separation was introduced.

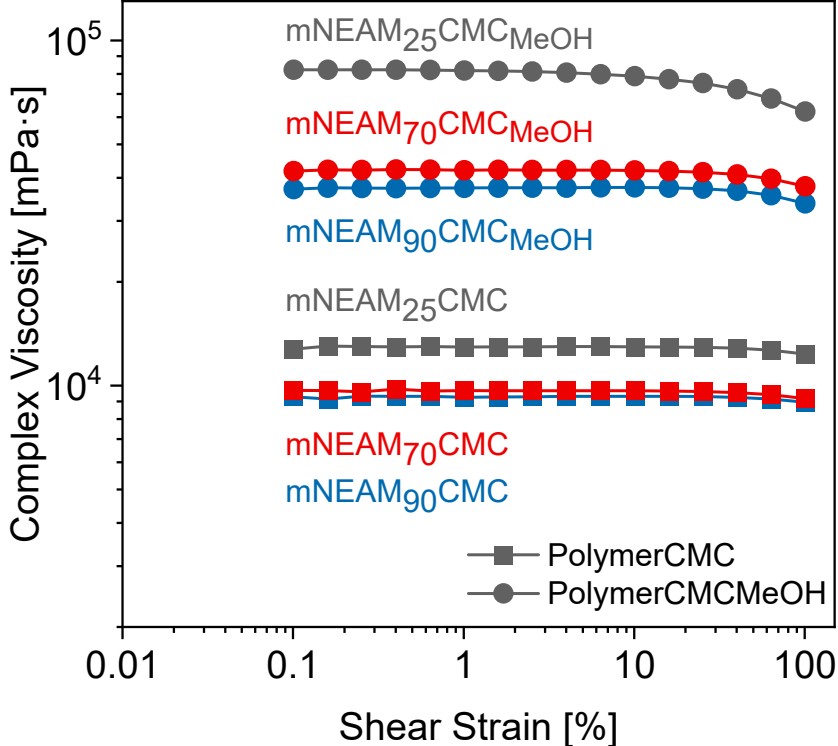

**Figure 6.** Complex viscosity as a function of shear strain (DSS) for the different blends at 3 °C.

## 4. Conclusions

New blends consisting of homopolymers (pNIPAM and pNEAM) and CMC were formed in water and 1:1 water/methanol mixtures. The molecular features were studied, and results were used for explaining the thermoresponsive behavior.

This work allowed us to analyze the competition between LCST, cononsolvency, and physical crosslinking through rheology.

In general, the introduction of CMC into the blends of pNIPAM and pNEAM homopolymers leads to a decrease in the LCST while increasing the pNEAM/pNIPAM ratio decreases the LCST moderately. No thermothickening behavior, typically observed for

CMC, is detected for the blends. This new behavior is probably promoted by pNEAM-chains, which could disrupt the hydrogen bonds performed between CMC and –OH groups. If the content of pNEAM-chains rises regarding pNIPAM-chains, the highest phase transition associated with LCST could decrease (>7 K) as interactions between chains rise, probably due to the collapse of the pNIPAM-chains.

Nevertheless, the samples performed in mixtures of methanol and water exclusively exhibit the pNEAM-LCST, while the pNIPAM-LCST is undetectable due to cononsolvency.

Rheology shows that high contents of hydrophilic pNEAM-chains could disrupt the hydrogen bonds between the CMC and methanol, negatively affecting the gelation compared to the samples with a higher proportion of pNIPAM chains where the gelation is clearly achieved. This fact explains that introducing hydrophilic homopolymers regarding pNIPAM and CMC could affect their hydrogen bonds, exhibiting a competition between physical-crosslinking and LCST. Then, these results suggest that introducing hydrophilic monomers could affect the interactions between the different polymeric matrices of the blends, opening a new perspective in this research field.

In comparison to our previous paper on p(NIPAM-co-NEAM) polymer blends with CMC [31], clearly, two distinct LCSTs were observed, while random-copolymers show only one LCST, linearly scaling with composition. Moreover, the increase of viscosity/modulus with increasing temperature is not observed for the blends, while it is obvious for the random-copolymers [31].

**Author Contributions:** Conceptualization, A.G.-P. and F.J.S.; methodology, A.G.-P. and F.J.S.; validation, F.J.S., S.H. and M.R.S.; formal analysis, A.G.-P.; investigation, A.G.-P. and W.L.; data curation, A.G.-P.; writing—original draft preparation, A.G.-P., W.L. and G.S.; writing—review and editing, F.J.S, S.H. and M.R.S.; supervision, F.J.S.; project administration, A.G.-P. and F.J.S.; funding acquisition, F.J.S. All authors have read and agreed to the published version of the manuscript.

**Funding:** The authors want to acknowledge the funding obtained from the National Science Foundation of China (21574086), Shenzhen Fundamental Research Funds (No. KC2014ZDZJ0001A), Shenzhen Sci & Tech research grant (ZDSYS201507141105130), and China Postdoctoral Science Foundation Grant (2018M633119).

**Institutional Review Board Statement:** Not applicable.

**Informed Consent Statement:** Not applicable.

**Data Availability Statement:** All raw data and evaluated data are available from the authors upon request.

**Conflicts of Interest:** The authors declare that they have no conflicts of interest. This article does not contain any studies involving animals performed by any of the authors. This article does not contain any studies involving human participants performed by any of the authors.

**Declarations:** Statements on compliance with ethical standards and standards of research involving humans and animals.

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
