# Peer review of "Hydrogen Bonds in Blends of Poly(N-isopropylacrylamide), Poly(N-ethylacrylamide) Homopolymers, and Carboxymethyl Cellulose"

_jcs, doi:10.3390/jcs5090240_

Round 1

Reviewer 1 Report

The manuscript by Stadler and co-workers deals with the synthesis of blends based on carboxymethylcellulose, poly(N-isopropylacrylamide) and/or poly(N-ethyl acrylamide). Beyond the blend synthesis, their physico-chemical characterization is also reported.
Unfortunately, the current work is too much similar to the following publication by the same Authors, also cited within the manuscript:
"Liang, W., García‐Peñas, A., Sharma, G., Kumar, A., & Stadler, F. J. (2020). Competition between Physical Cross‐Linking and Phase Transition Temperature in Blends Based on Poly (N‐isopropylacrylamide‐co‐N‐ethylacrylamide) Copolymers and Carboxymethyl Cellulose. Macromolecular Chemistry and Physics, 221(14), 2000081."

Several figures clearly overlap to those already published (i.e. Fig.1, Fig.3, Fig.5, ). Unless the Authors highlight the novelty of the current work, in this Reviewer opinion it seems too similar to the mentioned publication.

Reviewer 2 Report

Reviewer’s report

Title: Hydrogen bonds in blends of poly(N-isopropylacrylamide), poly(N-ethylacrylamide) homopolymers, and carboxymethyl cellulose.

Comments and Suggestions for Authors

Dear Authors,

General remarks: Firstly I would like to write, the manuscript is really very interesting, in my opinion the suitable use of the hydrogen bonds in polymer blends can be one of the promising and hot topics in the discovery of new materials properties in the near future. The current manuscript can be accepted after considering the following comments:

Introduction section: Please list a few unresolved problems related to the influence of hydrogen bonds on the change of the critical temperature of polymer blends. It is important in order to justify the research undertaken and it is in this part of the manuscript that it would be worth mentioning them. I would ask Authors to briefly indicate whether similar studies have been carried out in the past. If so, what new results could be expected from the research presented in the manuscript. Generally it is my request to show the novelty of the research.

In the section entitled: “Preparation of blends based on thermoresponsive polymers and CMC”: The first sentence of this chapter mentions two compounds poly (N-isopropylacrylamide), poly (N-ethyl acrylamide) chemical, I suggest including abbreviations of these compounds as well.

Was any statistical analysis of the obtained results carried out?

Line 286-290: The opaque blends composed of polymers and CMC prepared in water-alcohol mixtures has been observed for a long time. I would ask Authors to justify why the authors missed measuring the whiteness   value during the study and an analysis of the color change of the solution. It is an indicator of the quick assessment of the quality of cross-linking of the mixture.

 Conclusions are in accordance with the objectives and results. Generally conclusions are done correctly adequate to research results. Results and Discussion is presented clearly and concisely with adequate tables, figures, etc.

So my decision is: Minor revision, which you should made according the Reviewer's indications.

Best regards

Reviewer

Reviewer 3 Report

  1. Authors used abbreviation that was not defined. Authors should avoid using abbreviations in this section of the article because it may make difficult for the potential reader to understand the content of this article.
  2. When synthesizing chemical compounds, authors should include reaction schemes. This form is more legible and understandable than the description in the text.
  3. „The molecular weight and polydispersity indices (PDI - the ratio of weight Mw and number average molar mass Mn) of pNIPAM and pNEAM homopolymers were estimated from GPC-data, whose number average molar masses were Mn=29000 g/mol (PDI=1.52) and 3500 g/mol (PDI=1.05), respectively, based on PS-standards.” Authors did not report what was the weight average molecular weight for pNIPAM and pNEAM? Theoretically, this can be calculated from the formula for polydispersity. Nevertheless, the appropriate values should be included in the text.
  4. In my opinion, the spectra in Figure 1 are practically not discussed, e.g. one sentence concerning the FTIR analysis is definitely not enough. Also, the same applies to discussions of 1H NMR spectra. Why is there only 1 GPC chromatogram in Figure 1a?
  5. Authors should also perform 13C NMR of their samples.
  6. Authors used the abbreviation pNIPAM, and at other times PNIPAM or pNIPAm. Authors should standardize it.
  7. Figure 4. For the three measurement points, authors draw the trend line as if this dependence were linear. In my opinion, these are too few measuring points for such a dependence. Scatter charts should be used instead of line charts.
  8. I think that the discussion of the results should relate more to the research of other scientists, and not only to the general well-known dependencies.
  9. Conclusion section should be supported by research results, not only general statements.
  10. English is acceptable but needs some adjustments.
  11. The References section should be adapted to the journal's requirements.

Round 2

Reviewer 1 Report

The Authors clarified the novelty of their work as compared with the reference article published in Macromolecular Chemistry and Physics. Although not deeply original, the revised paper has been enriched with details and is now clearly distinguishable from the previous one.

Reviewer 3 Report

Manuscript has been significantly improved. It can be published in JCS.